# Multi-modal Denoising Diffusion Pre-training for Whole-Slide Image Classification

Wei Lou
The Chinese University of Hong Kong, Shenzhen
Shenzhen, China
Shenzhen Research Institute of Big Data
Shenzhen, China
weilou@link.cuhk.edu.cn

Guanbin Li
School of Computer Science and Engineering,
Sun Yat-sen University
Guangzhou, China
liguanbin@mail.sysu.edu.cn

Xiang Wan
Shenzhen Research Institute of Big Data
Shenzhen, China
Guangdong Provincial Key Laboratory of Big Data
Computing
Shenzhen, China
wanxiang@sribd.cn

Haofeng Li*
Shenzhen Research Institute of Big Data
Shenzhen, China
lhaof@sribd.cn

## Abstract

Whole-slide image (WSI) classification methods play a crucial role in tumor diagnosis. Most of them use hematoxylin and eosin (H&E) stained images, while Immunohistochemistry (IHC) staining provides molecular markers and protein expression information that highlights cancer regions. However, obtaining IHC-stained images requires higher costs in practice. In this work, we propose a multi-modal denoising diffusion pre-training framework that harnesses the advantages of IHC staining to learn visual representations. The framework is trained with the H&E-to-IHC re-staining task and IHC-stained image reconstruction task, which helps capture the structural similarity and staining difference between two image modalities. The trained model can then provide IHC-guided features, by taking only H&E-stained images as inputs. Besides, we build a new class-constraint contrastive loss to achieve the semantic consistency between dual-modal features from our pre-training framework. To integrate with WSI classifiers based on multi-instance learning, we further propose a bag feature augmentation strategy to extend bags with the features extracted by our pre-trained model. Experimental results on three datasets show that our pre-training framework effectively improves WSI classification and surpasses the state-of-the-art pre-training approaches. Code and model are released via https://github.com/lhaof/MDDP

## CCS Concepts

• **Computing methodologies** → **Image representations**.

*Corresponding author: Haofeng Li

## Keywords

Representation learning, Diffusion model, Image-to-image translation, Whole-slide image classification

**ACM Reference Format:**
Wei Lou, Guanbin Li, Xiang Wan, and Haofeng Li. 2024. Multi-modal Denoising Diffusion Pre-training for Whole-Slide Image Classification. In *Proceedings of the 32nd ACM International Conference on Multimedia (MM '24), October 28-November 1, 2024, Melbourne, VIC, Australia.* ACM, New York, NY, USA, 10 pages. https://doi.org/10.1145/3664647.3680882

## 1 Introduction

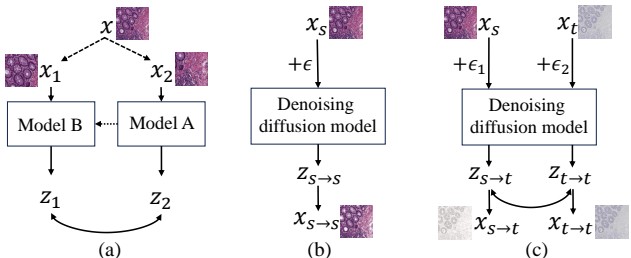

**Figure 1: (a) Typical contrastive learning based pre-training framework.** $x_1$ **and** $x_2$ **are different views of input image** $x$. **(b) Denoising diffusion pre-training on image reconstruction task.** $\epsilon$ **is the added noise.** $z_{t \rightarrow t}$ / $x_{s \rightarrow s}$ **denotes the reconstructed feature / image. (c) The proposed multi-modal denoising diffusion pre-training framework.** $z_{s \rightarrow t}$ **and** $x_{s \rightarrow t}$ **denote the re-stained feature and image generated in the re-staining task, respectively.** $z_{t \rightarrow t}$ / $x_{t \rightarrow t}$ **is the reconstructed feature / image in the reconstruction task.**

Histopathology images are widely used for medical diagnosis and research, facilitating objective assessments of diseases. Whole-slide images (WSIs) are high-resolution digital histopathology images

that visualize an entire tissue section. The creation of WSIs involves tissue fixation, cleaning, infiltration, sectioning, staining, and digital scanning. WSI classification including cancer classification [14, 50], grading [45, 55] and subtyping [7, 16], are important in tumor diagnosis and treatment planning. Hematoxylin and eosin (H&E) staining is widely used to visualize WSIs for its simplicity and affordability. However, H&E staining mainly emphasizes tissue structures and cell morphology, and its capability to detect specific cells and protein markers is limited [1, 62]. Hence, the assessment accuracy of disease through H&E-stained images is limited [30, 35, 62]. Immunohistochemistry (IHC) staining [10, 26, 27] labels specific antigens or proteins, offering molecular markers and protein expression levels for disease diagnosis. However, IHC-stained slides require extra labor, time, and specialized laboratory equipment for preparation [30, 62]. Therefore, it is an interesting challenge to leverage multi-modal training images to infer the labels of WSIs with some missing modality.

Some studies [11, 30, 35, 62] generate IHC-stained images from H&E-stained images using image-to-image translation. These methods typically employ generative adversarial networks (GANs) on paired H&E-IHC stained images to create virtual IHC-stained samples. However, their focus is limited to image generation and does not incorporate re-stained images or generated features to improve the performance of WSI analysis. Most WSI classification methods [37, 48, 58, 63] employ multi-instance learning (MIL) by dividing WSIs into patches and extracting patch features, which are then combined into a *bag*. Feature selection and classification are performed on the bag to obtain the category of the bag as the predicted WSI label. These methods are based solely on a single H&E staining modality, and few approaches have exploited the knowledge from multiple image modalities (such as IHC-stained images).

Self-supervised pre-training is widely used in learning visual representation, providing high-quality image features for downstream tasks. Masked image modeling [17, 21, 53, 57] recovers missing parts of input images for feature learning. Some methods based on contrastive learning [3, 5, 18, 32] learn to draw the features of positive samples (such as different views of an image) closer to each other (Figure 1(a)). Recent studies discovered that denoising diffusion models exhibit visual representation capabilities by estimating the added noise on input noisy images (Figure 1(b)) [6, 54, 56]. However, most diffusion-based pre-training methods mainly focus on the reconstruction task, overlooking the potential visual representation from multi-modal generation tasks, such as cross-modal image-to-image translation.

Inspired by the aforementioned observations, we propose a novel multi-modal denoising diffusion pre-training framework for whole-slide image classification (Figure 1(c)). First, to address the high acquisition costs of IHC-stained images, we introduce a re-staining task that generates IHC-modality images from H&E-modality inputs using a denoising diffusion model. Through the learning of shared structural features between H&E and IHC stained images and capturing the differences in staining appearance, the re-staining task enables the trained model to extract cross-modal visual representations. These representations can serve as supplementary multi-modal information for WSI analysis models in the absence of the IHC modality. Second, to enhance the generalization ability of the model and enable it to capture visual patterns across

different modalities, we take IHC-stained images as input during the training to tune the model on an extra IHC-stained image reconstruction task. The reconstruction task helps the framework to harvest specific antigen and protein expression features associated with IHC staining. To ensure semantic consistency between reconstructed and re-stained features, we propose to apply a contrastive loss between the features of the above two generative tasks. Furthermore, we observe that the presence of brown regions in IHC-stained samples correlates with the expression level of specific antibodies as well as the type of WSIs. Therefore, we can estimate binary pseudo-labels for each IHC-stained sample by quantifying the proportion of brown regions. To better guide the contrastive learning, we propose a class-constraint contrastive loss to encourage similarity between re-stained images and reconstructed images with the same pseudo labels, while ensuring lower feature similarity across different pseudo labels. The pseudo label based constraint is promising to preserve the similar clinical information between the reconstructed and re-stained representations.

For applying the multi-modal pre-trained model to WSI analysis, we develop a MIL-based WSI classification framework that is integrated with a new bag feature augmentation strategy. Within this MIL framework, we initially extract features from WSI patches using an ImageNet-trained encoder, forming a bag representation. Subsequently, we augment the bag with the features obtained from our pre-trained diffusion model. By doing so, we maintain the generality of the ImageNet pre-trained features while incorporating the IHC-guided features learned by our model.

To summarize, our main contributions have four folds:

- A multi-modal denoising diffusion pre-training framework that integrates image re-staining and reconstruction tasks for representation learning of histopathology images.
- A class-constraint contrastive loss that uses the prior image-level labels estimated from IHC-stained images to align global semantics of the two generative tasks..
- A bag feature augmentation strategy that equips existing multi-instance learning based WSI classifier with the features from our multi-modal pre-training framework.
- Experimental results indicate that our method not only boosts the classification performance of existing WSI classifiers but also achieves the state-of-the-art performance among existing pre-training algorithms.

## 2 Related Work

Most whole-slide image (WSI) classification methods [13, 28, 33, 37, 49, 63] are based on multiple-instance learning, which can be further divided into two groups: 1. using instance-level predictions directly, 2. using aggregation of instance-level features for bag-level classification. In the former group, Top-$K$ MIL [8] employs the top-$K$ instances for bag prediction. For the latter group, AB-MIL [23] calculates attention scores for all instances and computes the weighted average of instance features to obtain bag-level representations. HAG-MIL [58] utilizes multi-magnification images of WSIs, extracting features from higher magnification levels and distilling them to lower magnification levels to obtain bag-level representations. However, all these methods primarily focus on visual representations from H&E-stained images, which may be

improved when adding more specialized information from other staining techniques.

Image-to-image translation is the process of establishing a mapping between two image domains [22, 36, 46, 60, 61, 67]. Isola et al. [24] proposed Pix2Pix, a conditional generative adversarial network (GAN) for image-to-image translation using paired samples. Zhu et al. [66] proposed CycleGAN, which uses two GANs and allows the models to learn from unpaired images. However, due to their adversarial mechanism, GAN-based methods often encounter training difficulties and mode collapse issues. Recently, denoising diffusion models have been applied to image generation and translation due to their exceptional generative capabilities [9, 20, 38, 64]. BBDM [29] utilizes the Brownian bridge process to generate noisy images that resemble the target domain, rather than generating pure noise. In the area of histopathology, re-staining is an image-to-image task that involves translating the staining on histopathology images while preserving tissue structures and cell shapes. Current re-staining methods [11, 30, 35, 62] mainly employ GAN-based approaches trained on paired or unpaired images with different staining. However, these methods primarily focus on generating samples and do not thoroughly study the features learnt from the re-staining task for downstream image understanding tasks.

Vision representation learning involves pre-training a neural network to generate features or initialize model weights for downstream tasks. One type of visual pre-training method learns features based on the image itself using pretext tasks such as predicting transformations [15], inpainting [31, 41] or recovering masked regions [17, 21, 57]. Another type of visual pre-training method is based on contrastive learning [5, 40, 52, 65]. These methods utilize different views of the same sample or different samples as contrastive components for feature discrimination. Recently, several studies [6, 39, 54, 56, 59] have investigated the features learnt from diffusion denoising models (DDMs). However, current methods mainly rely on uni-modal image reconstruction tasks for denoising diffusion model pre-training, and it remains unclear whether cross-modal generative tasks can further enhance the representation ability of DDMs.

## 3 Methodology

### 3.1 Multi-modal Image Registration and Label Assignment.

As shown in Figure 2 (a), in our data acquisition process, an unstained tissue section is initially stained with H&E, and then stained with IHC after removing the H&E stains. This sequential staining approach ensures the maximum preservation of the positional and structural correspondence between the two images with different stains. However, the staining process may still undergo unavoidable transformations such as translation or rotation in digitization, resulting in misalignment. To address this, we first employ a rigid registration method [42] to align the H&E and IHC stained WSIs from the same tissue slide. Subsequently, we use a technique called Yottixel [25] to select the valid image patches from the registered H&E-stained WSIs based on the tissue segmentation results and RGB histograms. In Yottixel, the square regions of size $1000 \times 1000$ are cropped from the segmented tissue areas and resized to $256 \times 256$ image patches. The corresponding IHC-stained regions are cropped

and resized at the same positions as their paired H&E-stained image patches. After the above patch selection procedure, a large number of paired H&E-IHC stained samples $(x^{he}, x^{ihc})$ are generated.

*3.1.1 Pseudo Label Assignment.* In IHC staining, specific protein expression is visualized as brown regions, while normal tissues tend to appear more white. For example, as the row of $x^{ihc}$ in Figure 2 (a) shows, the left patch is a 'negative' sample and the right one with larger brown regions is a 'positive' sample. Therefore, the presence of specific protein expression can be evaluated by analyzing the proportion of brown areas in an IHC sample [62]. In our method, we define samples with brown regions larger than 1% as positive and those with less than 0.1% as negative, discarding the remaining samples. Each H&E-IHC pair is assigned a binary pseudo label $\rho$ (positive/negative) indicating the level of protein expression.

### 3.2 Multi-modal Denoising Diffusion Pre-training Framework

The proposed multi-modal denoising diffusion pre-training framework learns to extract IHC-staining guided representation with only H&E-stained images as inputs. The framework is trained with two tasks. The first one is a multi-modal image-to-image translation task, where the denoising diffusion model learns to map H&E-stained images to their IHC-stained counterparts. The second one is a uni-modal image reconstruction task, requiring the denoising diffusion model to recover IHC-stained images from their noisy version. As Figure 2 (b) shows, given a pair of H&E-stained and IHC-stained patch $x^{he}$ and $x^{ihc}$, we first employ a pre-trained encoder $\tau$ to extract their latent features $z_0^{he}$ and $z_0^{ihc}$. Then, we add noise to these latent features separately, resulting in the noisy features $z_t^{he}$ and $z_t^{ihc}$. Subsequently, a U-Net takes $z_t^{he}$ and $z_t^{ihc}$ as inputs to perform the re-staining and reconstruction tasks, respectively. Depending on the input features, the denoising U-Net model not only converts the latent features of H&E-stained patches into those of IHC-stained patches but also recovers the original features of IHC-stained patches from the noisy features. Let $f_t$ denotes the generated features of the image-to-image translation task, we could use a pre-trained decoder $\mathcal{D}$ to output synthetic IHC-stained images by using $f_t$ as input.

*3.2.1 Latent Feature Extraction.* We describe how to obtain the above-mentioned pre-trained encoder and decoder in the following. Latent diffusion models (LDMs) [2, 43, 47] move the diffusion process into latent space and perform denoising diffusion process on latent features, which effectively reduces the time and memory cost. In particular, LDMs utilize an autoencoder architecture comprising an encoder and a decoder. The encoder $\tau$ maps an input image $x$ to a latent feature map $z = \tau(x)$, while the decoder $\mathcal{D}$ converts the latent feature map back into an image. Prior to training the denoising U-net, we first train an autoencoder named VQGAN [12] on the H&E-IHC stained dataset. Then, during the training of our proposed framework, the weights of encoder $\tau$ and decoder $\mathcal{D}$ are frozen. The encoder takes paired H&E and IHC stained patches $(x^{he}$ and $x^{ihc})$ with the shape of $h \times w \times 3$ as inputs and outputs two latent features $Z_0^{he}$ and $Z_0^{ihc}$ of shape $h' \times w' \times c'$. The two latent features will be used in the diffusion process.

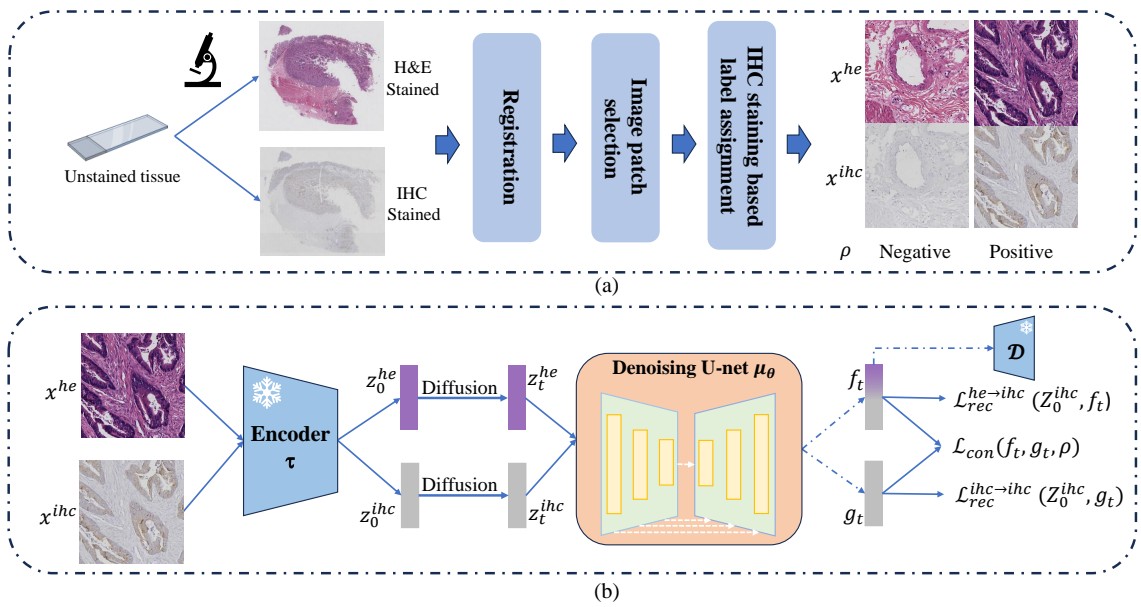

**Figure 2: (a) Multi-modal image registration and label assignment. The IHC staining based label assignment is to assign a positive/negative pseudo label for each H&E-IHC stained patch pair based on the percentage of brown regions in the IHC-stained patch. (b) Multi-modal denoising diffusion pre-training framework. We employ a denoising-diffusion model to accomplish the re-staining task and the reconstruction task on two staining modalities at the feature level. $f_t$ represents the generated features of the re-staining task, while $g_t$ represents the generated features of the reconstruction task.**

*3.2.2 Diffusion Process.* The diffusion process in our framework introduces noise to input features in each time step $t$, resulting in noisy features after a long time. Our framework follows an image-to-image translation network called BBDM [29] to perform diffusion processes. In the diffusion process, noise is added to two latent features for different tasks. The forward diffusion process of the two tasks can be defined as:

$$q_{he->ihc}\left(z_t^{he} \mid z_0^{he}, z_0^{ihc}\right) = \mathcal{N}\left(z_t^{he}; (1-m_t)z_0^{he} + m_t z_0^{ihc}, \delta_t I\right),$$
(1)

$$q_{ihc->ihc}\left(z_t^{ihc} \mid z_0^{ihc}\right) = \mathcal{N}\left(z_t^{ihc}; z_0^{ihc}, \delta_t I\right),$$
(2)

where $q_{he->ihc}$ represents the forward transition probability for the H&E feature to IHC feature translation task, and $q_{ihc->ihc}$ denotes the forward transition probability for the IHC feature reconstruction task. $\mathcal{N}$ is the Gaussian distribution. $t$ is a time step between 0 and $T$. $m_t = t/T$ denotes the proportion of the added noise. $\delta_t$ is designed as $2(m_t - m_t^2)$ following the best setting of BBDM [29]. In the $t^{th}$ time step, the diffused features for two tasks are computed as:

$$z_t^{he} = (1-m_t)z_0^{he} + m_t z_0^{ihc} + \sqrt{\delta_t}\epsilon_t,$$
(3)

$$z_t^{ihc} = z_0^{ihc} + \sqrt{\delta_t}\epsilon_t,$$
(4)

where $z_t^{he}$ and $z_t^{ihc}$ denote the features of H&E-stained and IHC-stained images after adding noises at time step $t$. $\epsilon_t \sim \mathcal{N}(0, I)$ is Gaussian noise from a standard normal distribution.

*3.2.3 Denoising Process.* In the denoising process, given $z_t^{he}$, $z_t^{ihc}$ and $t$, a U-net model $\mu_\theta$ is deployed to predict the noise that is added on these two input noisy features respectively. It is important to notice that $\mu_\theta$ learns to predict noise for different tasks (image-to-image translation/image reconstruction) purely relying on the input features. The denoised features can be calculated as $f_t = z_t^{he} - \mu_\theta(z_t^{he}, t)$ and $g_t = z_t^{ihc} - \mu_\theta(z_t^{ihc}, t)$. $f_t \in \mathbb{R}^{h' \times w' \times c'}$ represents a synthetic IHC virtual-stained feature map from the noisy feature map of its paired H&E-stained image patch. $g_t \in \mathbb{R}^{h' \times w' \times c'}$ denotes the reconstructed feature map of the input noisy feature map of the original IHC-stained patch.

## 3.3 Training Scheme

*3.3.1 Reconstruction and Re-staining losses.* In our proposed multi-modal pre-training network, there are two generation losses that measure the synthesis quality for reconstruction and re-staining tasks, respectively. One is to evaluate whether the denoised IHC features generated through a reconstruction task are consistent with their original features extracted by the encoder $\tau$. This reconstruction loss helps the denoising U-net better learn the distribution of the original features of IHC-stained slides. The other generation loss is to assess whether the features converted by the denoising diffusion model using H&E-stained image patches through a re-staining task are similar to features of their corresponding IHC-stained patches. This loss enables the denoising U-Net to predict cross-modal IHC-staining guided representations with uni-modal H&E-stained inputs. We use L1 loss for generation losses, and they are defined as $\mathcal{L}_{rec}^{he \to ihc} = ||Z_0^{ihc} - f_t||$ and $\mathcal{L}_{rec}^{ihc \to ihc} = ||Z_0^{ihc} - g_t||$,

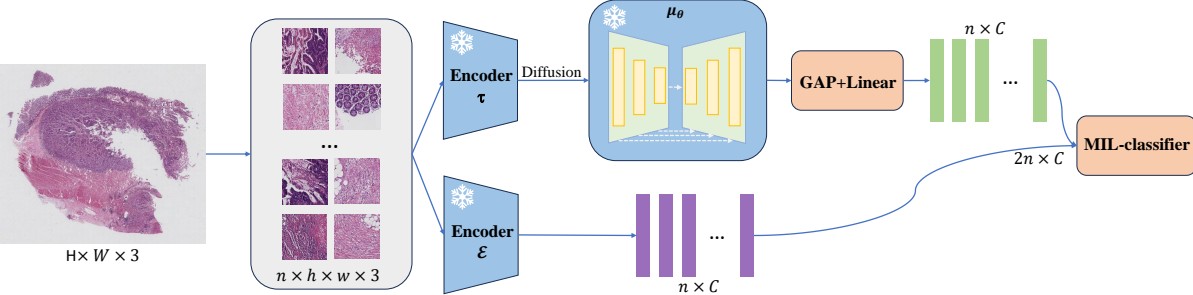

**Figure 3: Bag feature augmentation for multi-instance learning based whole-slide image classification. The encoder $\tau$, a single-step diffusion process and the U-net $\mu_\theta$ from our pre-training framework are integrated into a feature extractor to generate visual representations for H&E-stained image patches. The weights of $\tau$ and $\mu_\theta$ are frozen during the training of MIL-based classifier. The encoder $\mathcal{E}$ is a ResNet pre-trained on ImageNet.**

where $\mathcal{L}_{rec}^{ihc \to ihc}$ and $\mathcal{L}_{rec}^{he \to ihc}$ denote the losses for reconstruction task and re-staining task, respectively.

*3.3.2 Class-constraint Contrastive Loss.* As mentioned in Section 3.1, each IHC-stained image patch can be categorized into 'positive' or 'negative' based on the percentage of brown regions. The H&E-stained image patches share the same categories with their paired IHC-stained image patches. Therefore, we propose a class-constraint contrastive loss to maintain the semantic consistency between re-stained features $f_t$ and reconstructed features $g_t$. For a batch of re-stained features $F = \{f_t^0, f_t^1, ..., f_t^b\}$ and a batch of corresponding reconstruction features $G = \{g_t^0, g_t^1, ..., g_t^b\}$, they share the same pseudo labels $\rho = \{\rho^0, \rho^1, ..., \rho^b\}$. The features of the same label should be similar in the latent space. Therefore, we define a CLIP [44] -style contrastive loss to perform class-constraint feature alignment:

$$\mathcal{L}_{con} = -\sum_{i=0}^{b} \left[ \rho^i \cdot \log \left( \frac{e^{(\phi(f_t^i) \cdot \phi(g_t^i)/\sigma)}}{\sum_{j \neq i} e^{(\phi(f_t^j) \cdot \phi(g_t^j)/\sigma)}} \right) \right.$$
$$\left. + (1 - \rho^i) \cdot \log \left( \frac{e^{(-\phi(f_t^i) \cdot \phi(g_t^i)/\sigma)}}{\sum_{j \neq i} e^{(-\phi(f_t^j) \cdot \phi(g_t^j)/\sigma)}} \right) \right], \quad (5)$$

where $b$ is the batch size. $f_t^i$ and $g_t^i$ denote the re-stained feature and reconstructed feature of the $i$th pair in an input batch, respectively. $f_t^j$ and $g_t^j$ represent the other features in the batch. $\phi$ is a global average pooling operation that converts all the features in $F$ and $G$ to feature vectors of shape $1 \times c'$. $\sigma$ is the temperature factor. The overall loss function has three terms:

$$\mathcal{L}(X^{he}, X^{ihc}, \rho) = \lambda_1 \mathcal{L}_{rec}^{he \to ihc} + \lambda_2 \mathcal{L}_{rec}^{ihc \to ihc} + \lambda_3 \mathcal{L}_{con}, \quad (6)$$

where $X^{he}, X^{ihc}$ and $\rho$ denote a batch of H&E-stained image patches, IHC-stained image patches and their correspoding pseudo labels, respectively. $\lambda_1, \lambda_2$ and $\lambda_3$ are set 10, 1 and 0.1, respectively. The study of hyper-parameters is in Supplementary Materials.

## 3.4 Bag Feature Augmentation for Downstream Tasks

For the pre-trained denoising diffusion network, we employ it as a feature extractor for H&E-stained images, rather than an image

generator. Considering that the denoising U-Net has learned to transform the features of H&E-stained images into the representations of IHC-stained images, we assume that it can provide not only the morphological and textural features of H&E-stained images but also the IHC-staining guided features indicating the level of specific protein expression.

As depicted in Figure 3, for the task of classifying whole-slide images (WSI), an H&E-stained WSI of shape $H \times W \times 3$ is cropped into $n$ image patches, which form a bag $\mathcal{B}$ for multi-instance learning. This bag is then sent into an ImageNet pre-trained encoder $\mathcal{E}$ to extract universal instance-level features $f_{uni} \in \mathbb{R}^{n \times C}$. Simultaneously, the bag is also input into the other feature extractor that is pre-trained using our proposed multi-modal framework, aiming to extract IHC-guided representations. The feature extractor consists of the pre-trained encoder $\tau$, a single-step diffusion process $\Psi$, and the denoising U-net $\mu_\theta$. For each image patch $x^{he}$ in $\mathcal{B}$, a feature map $z_0^{he}$ is initially computed using $\tau$. Then, given a time step $t$, a noised feature $z_t^{he}$ can be obtained through a single-step diffusion process: $z_t^{he} = z_0^{he} + \sqrt{\delta_t}\epsilon_t$, where $\delta_t$ and $\epsilon_t$ are the same as those in Eq. (3). The difference between Eq. (3) and the single-step diffusion process is that $z_0^{ihc}$ in Eq. (3) is replaced by $z_0^{he}$ since we aim to solve the downstream tasks with only uni-modal H&E-stained images. Subsequently, the denoising U-net model takes the noised features and time step $t$ as input, and outputs feature representations $f_{de} \in \mathbb{R}^{n \times h'' \times w'' \times c''}$.

To align features of different latent spaces from two pre-trained models, we apply a global average pooling operator and a linear layer (see 'GAP+Linear' in Figure 3) to the features of our pre-trained model. Then, we concatenate $f_{de}$ and $f_{uni}$ to build the augmented bag-level features $f_{bag} \in \mathbb{R}^{2n \times C}$. The extraction of the bag-level features can be formulated as follows:

$$f_{bag} = FC(\phi(\mu_\theta(\Psi(\tau(\mathcal{B}), t), t)))||\mathcal{E}(\mathcal{B}), \quad (7)$$

where $\phi$ denotes the global average pooling operation and $FC$ is the linear layer (Fully-Connected layer). $\tau, \Psi$ and $\mu_\theta$ represent the pre-trained encoder, the single-step diffusion process and the denoising U-Net in our proposed multi-modal framework, respectively. $\mathcal{E}$ represents the ImageNet pre-trained encoder. || means the concatenating operation. Then, the resulting bag-level features can be

**Table 1: Comparison between existing WSI classification approaches without and with our pre-trained features (+Ours) on three datasets. The subscript in each cell is the standard derivation.**

| Methods | Camelyon16 | | | TCGA-COAD | | | TCGA-NSCLC | | |
|---|---|---|---|---|---|---|---|---|---|
| | AUC | F1 | Acc | AUC | F1 | Acc | AUC | F1 | Acc |
| CLAM-SB | $0.831_{0.051}$ | $0.762_{0.055}$ | $0.852_{0.028}$ | $0.881_{0.035}$ | $0.620_{0.074}$ | $0.874_{0.058}$ | $0.903_{0.022}$ | $0.833_{0.031}$ | $0.823_{0.011}$ |
| CLAM-SB + Ours | $0.868_{0.043}$ | $0.793_{0.042}$ | $0.872_{0.033}$ | $\mathbf{0.898}_{0.031}$ | $\mathbf{0.710}_{0.045}$ | $\mathbf{0.918}_{0.028}$ | $0.925_{0.042}$ | $0.850_{0.021}$ | $0.841_{0.033}$ |
| CLAM-MB | $0.848_{0.046}$ | $0.757_{0.055}$ | $0.841_{0.028}$ | $0.853_{0.051}$ | $0.534_{0.088}$ | $0.879_{0.013}$ | $0.925_{0.019}$ | $0.853_{0.021}$ | $0.851_{0.024}$ |
| CLAM-MB + Ours | $0.872_{0.047}$ | $0.786_{0.052}$ | $0.857_{0.031}$ | $0.882_{0.047}$ | $0.650_{0.063}$ | $0.902_{0.021}$ | $0.941_{0.026}$ | $0.878_{0.043}$ | $0.875_{0.046}$ |
| DTFD-MIL | $0.923_{0.032}$ | $0.840_{0.028}$ | $0.874_{0.021}$ | $0.851_{0.025}$ | $0.530_{0.077}$ | $0.854_{0.055}$ | $0.902_{0.021}$ | $0.831_{0.032}$ | $0.847_{0.043}$ |
| DTFD-MIL + Ours | $0.934_{0.023}$ | $0.863_{0.010}$ | $0.905_{0.020}$ | $0.863_{0.020}$ | $0.674_{0.043}$ | $0.890_{0.010}$ | $0.922_{0.030}$ | $0.843_{0.033}$ | $0.857_{0.031}$ |
| HAG-MIL | $0.936_{0.012}$ | $0.887_{0.021}$ | $0.880_{0.011}$ | $0.762_{0.013}$ | $0.645_{0.011}$ | $0.883_{0.017}$ | $0.940_{0.022}$ | $0.867_{0.034}$ | $0.867_{0.034}$ |
| HAG-MIL + Ours | $\mathbf{0.953}_{0.013}$ | $\mathbf{0.905}_{0.022}$ | $\mathbf{0.893}_{0.023}$ | $0.802_{0.052}$ | $0.703_{0.044}$ | $0.898_{0.063}$ | $\mathbf{0.951}_{0.017}$ | $\mathbf{0.881}_{0.020}$ | $\mathbf{0.883}_{0.020}$ |

employed in a wide range of MIL-based WSI classifiers [37, 58, 63] to obtain the final predictions.

The denoising U-Net model offers multi-scale feature representations. Previous studies [6, 56] have shown that utilizing features from different decoder layers can affect the performance of downstream tasks. Hence, we conducted experiments to study how to select the decoder layer of the denoising U-Net to extract features. We select the optimal feature from these layers to be the final output representations of our pre-trained model. Besides, the different choices of $t$ can also affect the strength of extracted features. Therefore, like former studies, we search for the best $t$ that achieves the superior performance for downstream tasks. The details are in the experimental section.

## 4  Experiments

### 4.1  Implementation Details

For pre-training, we built an **In-house paired H&E-IHC stained dataset**, which has 184 pairs of H&E-stained and IHC-stained WSIs of Colon cancer from a local hospital. After registration, we extracted 13,248 pairs of image patches at a magnification of 20x. Each image patch was cropped and resized to $256 \times 256$. Subsequently, we divided the dataset into training and validation sets in an 8:2 ratio for pre-training. For downstream tasks, we use the following dataset. **Camelyon16** [34] is a public WSI classification dataset of breast cancer, including 270 training WSIs and 129 testing WSIs. The training set contains 159 normal WSIs and 111 tumor WSIs. For each method, we perform 10-fold cross-validation on the training set of Camelyon16 to obtain 10 models of different weights, and average their results on the official testing set. **TCGA-COAD** is a WSI subtyping dataset of Colon cancer from the TCGA database [51], collected by us. The dataset consists of 392 WSIs categorized as adenomas and adenocarcinomas (adenomas) and 65 WSIs categorized as cystic, mucinous, and neoplasms (non-adenomas). We further divided the dataset into training, validation, and test sets in a ratio of 6:1.5:2.5. 5-fold cross-validation is adopted, and the mean value of performance metrics of the 5 testing folds are reported. **TCGA-NSCLC** is a public WSI subtyping dataset from the TCGA database [51], and has two kinds of Lung tumor: 541 Lung Squamous Cell Carcinoma WSIs (TGCA-LUSC) and 512 Lung Adenocarcinoma WSIs (TCGA-LUAD). We split the dataset into

training, validation, and test sets by a ratio of 6:1.5:2.5. For evaluation, we employ 5-fold cross-validation and report the average result across the 5 testing folds.

We use AUC, F1 score, and accuracy (Acc) as evaluation metrics. The thresholds for F1 score and accuracy are 0.5 [37, 58, 63]. To train our framework, the number of time steps $T$, the batch size, the number of epochs are set to 1000, 96, and 100, respectively. Adam optimizer is used with a learning rate of $1 \times 10^{-4}$. For WSI classification, the Encoder $\mathcal{E}$ in the bag feature augmentation is ResNet-50 [19]. We extract 9 feature maps ($u_1$-$u_9$) from the denoising U-net $\mu_\theta$ and the synthetic IHC virtual-stained feature map $f_t$ to search for the optimal feature. The sizes of these extracted feature maps are $1024 \times 4 \times 4$ ($u_1$-$u_3$), $512 \times 8 \times 8$ ($u_4$-$u_6$), $128 \times 16 \times 16$ ($u_7$-$u_9$), and $256 \times 16 \times 16$ ($f_t$). For the Camelyon16 / TCGA-COAD / TCGA-NSCLC dataset, we adopt the feature map $u_4$ / $u_3$ / $u_5$ and a time step of 40 / 20 / 20.

### 4.2  Comparison with Existing Methods

*4.2.1  Comparison with previous WSI classification models.* We study if the features from our pre-training framework can enhance existing WSI classifiers with the proposed bag feature augmentation strategy. In Table 1, we evaluate four WSI classifiers (CLAM-SB [37], CLAM-MB [37], DTFD-MIL [63], and HAG-MIL [58]), and compare their results without using and with our pre-trained features (+Ours) on three datasets. The approaches without '+Ours' only use ImageNet pre-trained features from a ResNet to achieve multi-instance learning as their original setting. As Table 1 presents, the best classification results on the Camelyon16, TCGA-COAD, and TCGA-NSCLC datasets are obtained by our pre-trained features and bag augmentation strategy with HAG-MIL, CLAM-SB, and HAG-MIL, respectively. Our learned features and augmentation strategy improve HAG-MIL by 1.8% F1 score on the Camelyon16 dataset, CLAM-SB by 9% F1 score on the TCGA-COAD dataset, and HAG-MIL by 1.6% accuracy on the TCGA-NSCLC dataset, respectively. Note that our proposed method consistently improves the four baseline models across three datasets and in all three metrics. The observed improvements range from 1.1% to 4% in AUC, 1.2% to 14.4% in F1 score, and 1.0% to 4.4% in accuracy. That shows the generalization ability of our method.

**Table 2: Comparison between the state-of-the-art visual pre-training approaches and our proposed framework. All the experiments use HAG-MIL [58] as the WSI classifier.**

| Methods | Camelyon16 | | | TCGA-COAD | | |
|---|---|---|---|---|---|---|
| | AUC | F1 | Acc | AUC | F1 | Acc |
| Baseline | $0.936_{0.01}$ | $0.887_{0.02}$ | $0.880_{0.01}$ | $0.762_{0.01}$ | $0.645_{0.01}$ | $0.883_{0.02}$ |
| DINO | $0.945_{0.03}$ | $0.883_{0.01}$ | $0.891_{0.03}$ | $0.788_{0.07}$ | $0.701_{0.08}$ | $0.879_{0.03}$ |
| Simmim | $0.935_{0.04}$ | $0.887_{0.03}$ | $0.878_{0.06}$ | $0.754_{0.08}$ | $0.621_{0.07}$ | $0.840_{0.09}$ |
| HIPT | $0.905_{0.03}$ | $0.832_{0.02}$ | $0.851_{0.03}$ | $0.772_{0.05}$ | $0.591_{0.10}$ | $0.849_{0.02}$ |
| DINOv2 | $0.951_{0.03}$ | $0.888_{0.06}$ | $0.880_{0.05}$ | $0.712_{0.09}$ | $0.619_{0.02}$ | $0.871_{0.09}$ |
| Ours | $\mathbf{0.953}_{0.01}$ | $\mathbf{0.905}_{0.02}$ | $\mathbf{0.893}_{0.02}$ | $\mathbf{0.802}_{0.05}$ | $\mathbf{0.703}_{0.04}$ | $\mathbf{0.898}_{0.06}$ |

*4.2.2 Comparison with the state-of-the-art pre-training methods.* In Table 2, our pre-training framework is compared with the state-of-the-art methods on the Camelyon16 and TCGA-COAD datasets. Among these methods, Simmim [57] is based on masked image modeling, while DINO [3], DINOv2 [40] and HIPT [4] are built on contrastive learning. HIPT is proposed for histopathology images. For fair comparison, we integrate their pre-trained encoder directly into our bag feature augmentation strategy, replacing our feature extractor. HAG-MIL is chosen as the WSI classifier. 'Baseline' is the HAG-MIL classifier that only uses ImageNet pre-trained features without our proposed bag augmentation strategy. For HIPT, we directly use its released weights trained on a large amount of histopathology images. For Simmim, DINO, and DINOv2, we compare their results using their ImageNet pre-trained weights and those using the weights re-trained on our in-house dataset and the training set of CAMELYON16 / TCGA-COAD, reporting the best results. Specifically, the reported results for Simmim and DINO are based on the re-trained models, while for DINOv2, the best result is based on its ImageNet pre-trained model. In Table 2, our framework surpasses the state-of-the-art pre-training methods in WSI classification. Specifically, on the Camelyon16 dataset, our framework achieves 1.7% higher F1 score compared to the second best DINOv2. For the TCGA-COAD dataset, our framework outperforms the second best DINO by 1.4% AUC and 1.9% accuracy.

## 4.3 Ablation Study

We study the effectiveness of the H&E to IHC re-staining task, the IHC-stained image reconstruction task, the class-constraint contrastive loss in our pre-training framework and the bag feature augmentation strategy. The results are obtained on the Camelyon16 dataset with CLAM-SB as the WSI classifier.

*4.3.1 Re-staining task:* In Table 3 (a), 'M1' denotes the WSI classifier takes only ImageNet pre-trained features as inputs. 'M2' means that our pre-training framework only performs H&E to IHC re-staining task. The comparison of 'M1' and 'M2' validates that the features learned from the re-staining task can enhance the WSI classification by 0.9% AUC on Camelyon16 dataset.

*4.3.2 Reconstruction task:* To show the strength of the IHC-stained image reconstruction task, a model 'M3' is built by removing the reconstruction task from our framework. In the training stage of 'M3', $g_t$ in $\mathcal{L}_{con}$ is not set to generated feature maps, but set to

**Table 3: Ablation study on the Camelyon16 dataset using CLAM-SB [37] as the WSI classifier.**

| $\mathcal{E}$ | $\mathcal{L}_{rec}^{he \rightarrow ihc}$ | $\mathcal{L}_{con}$ | $\mathcal{L}_{rec}^{ihc \rightarrow ihc}$ | Camelyon16 | | |
|---|---|---|---|---|---|---|
| | | | | AUC | F1 | Acc |
| M1 | ✓ | | | | $0.831_{0.05}$ | $0.762_{0.06}$ | $0.852_{0.03}$ |
| M2 | ✓ | ✓ | | | $0.840_{0.03}$ | $0.770_{0.03}$ | $0.859_{0.01}$ |
| M3 | ✓ | ✓ | ✓ | | $0.861_{0.04}$ | $0.782_{0.05}$ | $0.868_{0.04}$ |
| M4 | ✓ | ✓ | | ✓ | $0.820_{0.08}$ | $0.753_{0.03}$ | $0.841_{0.07}$ |
| M5 | ✓ | ✓ | ✓ | ✓ | $\mathbf{0.868}_{0.04}$ | $\mathbf{0.793}_{0.04}$ | $\mathbf{0.872}_{0.03}$ |

(a)

| Methods | Camelyon16 | | |
|---|---|---|---|
| | AUC | F1 | Acc |
| Ours without $\rho$ | $0.837_{0.01}$ | $0.781_{0.05}$ | $0.858_{0.04}$ |
| Ours with $\rho$ | $\mathbf{0.868}_{0.04}$ | $\mathbf{0.793}_{0.04}$ | $\mathbf{0.872}_{0.03}$ |

(b)

$z_0^{ihc}$ that is extracted by the encoder $\tau$ using IHC-stained images. By comparing 'M3' and 'M5', we show that the IHC-stained image reconstruction task enhances the pre-trained features and increases the WSI classification results by 0.7% AUC and 1.1% F1 score.

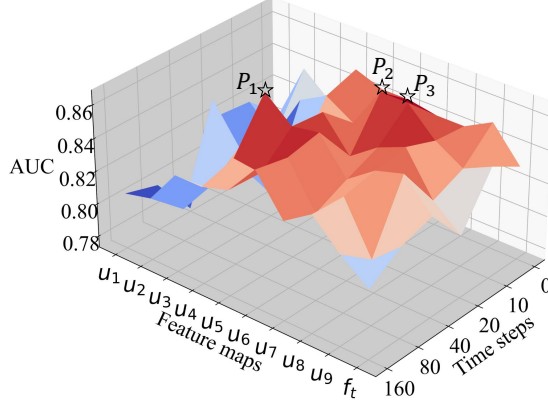

**Figure 4: Hyper-parameter investigation of different time steps and different feature maps on the Camelyon16 dataset using CLAM-SB[37] as the classifier.**

*4.3.3 Class-constraint contrastive loss:* In Table 3 (a), we present the effectiveness of the proposed class-constraint contrastive loss by comparing the models from 'M2' to 'M5'. First, comparing 'M2' to 'M3', the class-constraint contrastive loss improves the features of our pre-training framework and increases the classification performance by 2.1% AUC, 1.2% F1 score, and 1.1% accuracy on Camelyon16 dataset. Besides, comparing 'M4' and 'M5' reveals that the proposed contrastive loss helps improve the semantic consistency between multi-modal representations, significantly enhancing the classification results by 4.8% AUC, 4.0% F1 score, and 3.1% accuracy. We further study the pseudo label $\rho$ in Table 3 (b). 'Ours without $\rho$' denotes the contrastive loss of our framework and is only applied to narrow the difference between paired H&E-IHC features without

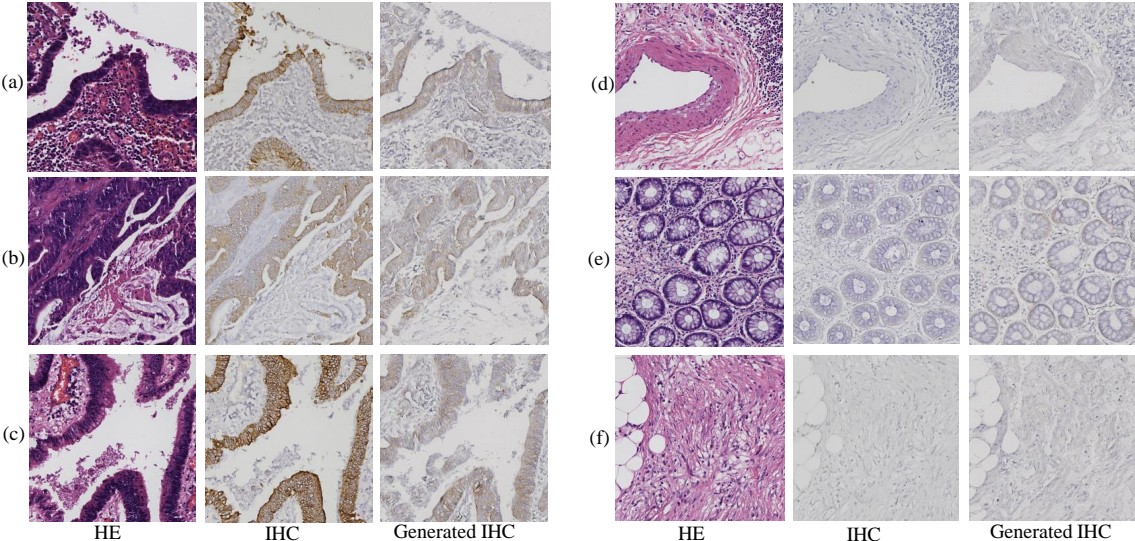

**Figure 5: Visualization of the generated IHC virtual-stained image patches.**

any constraints from the pseudo labels. The results show that the pseudo label constraints can enhance the feature representations and boost the classification performance by 3.1% AUC, 1.2% F1 score, and 1.4% accuracy.

*4.3.4 Bag feature augmentation:* Comparing 'M1' to 'M2', 'M3', and 'M5' in Table 3 (a) shows that if high-quality representations are learned, our bag feature augmentation strategy can bring in substantial improvements in the performance of WSI classification.

## 4.4 Hyperparameter Investigation

We investigate how different extracted feature maps and various time steps ($t$) in our pre-trained model affect the WSI classification. To select the optimal feature maps, we evaluate the classification performance using each of the multi-scale feature maps ($u_1 - u_9$) from the decoder of the denoising U-net and the synthetic IHC virtual-stained feature map $f_t$. In Figure 4, the height of each point on the 3D mesh represents the AUC value of a WSI classifier trained based on a specific feature map and time step $t$. We observe that the feature maps from intermediate decoder layers ($u_4 - u_8$) are more likely to achieve better performance, exhibiting improved stability and reduced variance. For time steps, we selected six different values (0,10,20,40,80,160) as the inputs to the single-step diffusion process and the denoising U-net. The results in Figure 4 suggest that the different choices of time step have limited influence on the representation quality for WSI classification task. There are three peaks in the 3D-mesh: P1 (u4,40,0.868), P2 (u6,10,0.857) and P3 (u7,10,0.857). For the three WSI classification datasets, we conduct experiments similar to Figure 4 to determine the feature map and time step on the validation sets.

## 4.5 Visual Results

Figure 5 shows the synthetic IHC-stained images generated by our pre-training framework in the re-staining task. Figure 5(a)-(c) display the H&E-IHC stained image pairs with pseudo labels as 'positive', along with the generated IHC virtual-stained images ('Generated IHC'). Figure 5(d)-(f) show the image pairs with pseudo labels as 'negative' and the generated IHC virtual-stained images. Note that the generated IHC images in (a)-(c) do show some darker regions that are consistent with the IHC images with positive labels. The results verify that our multi-modal pre-training framework can well capture the IHC-related information using only H&E images as inputs and generate corresponding brown/white IHC-stained regions.

## 5 Conclusion

In this paper, we present a novel multi-modal denoising diffusion pre-training framework for solving the task of histopathology image analysis. Firstly, we train a denoising diffusion model on the H&E-to-IHC image translation task and IHC-stained image reconstruction task, allowing the model to provide multi-modal information from even an image modality. Next, we introduce a class-constraint contrastive loss that ensures semantic consistency between the re-stained features and the reconstructed ones, utilizing prior pseudo labels estimated from IHC-stained images. Moreover, to integrate our synthetic multi-modal features with existing MIL-based WSI classifiers, we propose a new bag feature augmentation strategy to expand bag features with the generated features from our pre-training framework. Experimental results demonstrate that our proposed framework effectively improves the performance of existing WSI classification methods and outperforms the state-of-the-art pre-training approaches.

# Acknowledgments

This work was supported in part by the Shenzhen Science and Technology Program (JCYJ20220818103001002), in part by the Guangdong Provincial Key Laboratory of Big Data Computing, The Chinese University of Hong Kong, Shenzhen, in part by the Longgang District Special Funds for Science and Technology Innovation (LGKCSDPT2023002), in part by the National Natural Science Foundation of China (NO. 62102267, NO. 62322608), in part by the Guangdong Basic and Applied Basic Research Foundation (2023A1515011464), in part by the Fundamental Research Funds for the Central Universities under Grant 22lgqb25.

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
