# OpenReview forum: "Multi-modal Denoising Diffusion Pretraining for Whole-Slide Image Classification"
_acmmm.org/ACMMM/2024/Conference — MM2024 Poster_

### Official Review · Reviewer_2nAu · 2024-05-04

**Rating:** 5
**Confidence:** 4

**Summary:**

The paper introduces a novel multi-modal denoising diffusion pre-training framework which leverages IHC staining for visual representation learning. Additionally, a class-constraint contrastive loss is introduced to ensure semantic coherence among dual-modal features. To integrate with WSI classifiers employing multi-instance learning, a bag feature augmentation strategy is designed to enrich the bags with features extracted by the proposed pre-trained model.

The paper demonstrates the effectiveness of the method by comparing it with current state-of-the-art methods across multiple datasets.

**Strengths:**

1. The method proposed in the paper is innovative. It is worth exploring to learn feature representations through the task of virtual staining from H&E images to IHC, as well as the reconstruction task of IHC images.

2. The paper is well-organized and easy to follow.

3. The experimental design of the paper is rigorous.

**Limitations:**

1. For the virtual staining from H&E to IHC, current methods seem unable to produce biologically meaningful IHC images. Even in the generated IHC images shown in Fig. 5 of the paper, there are numerous discrepancies compared to the corresponding ground truth IHC images. Is there any value in learning feature representation from such an "incorrect staining transformation"?

2. When creating the dataset, did you evaluate the extent of tissue sample damage caused by first using H&E staining and then washing off the dye before applying IHC staining?

3. In Sec. 4.4, do you mean "we evaluate the classification performance using each of the multi-scale feature maps from the decoder of the denoising U-net $\textbf{as}$ the synthetic IHC virtual-stained feature map"?

I believe this is a very meaningful and systematic work. Are you planning to open-source the code?

**Suitability:**

3

---

### Official Review · Reviewer_EkUT · 2024-05-17

**Rating:** 3
**Confidence:** 2

**Summary:**

In this paper, the authors introduce a pretraining method for WSI data. Specifically, they employ a denoising diffusion model coupled with a contrastive loss, which guarantees semantic consistency between the retained and reconstructed features. Experimental results indicate that the proposed method outperforms existing pre-training models, thus demonstrating its effectiveness in handling WSI data.

**Strengths:**

- **S1:** The research on WSI pre-training  is a trending direction.

- **S2:** The method proposed by the author achieves improved results across multiple datasets and against various baselines.

**Limitations:**

- **W1:**  The proposed method appears to be a straightforward combination of dilution and contrastive techniques.
- **W2:** The author has introduced an excessive number of hyperparameters, which may complicate the training and deployment of the model.

**Suitability:**

2

---

### Official Review · Reviewer_Q4zY · 2024-05-24

**Rating:** 3
**Confidence:** 3

**Summary:**

In this paper, a multimodal denoising diffusion pre-training framework is introduced that leverages IHC staining to learn visual representations. It involves training with H&E-to-IHC re-staining and IHC image reconstruction tasks, enhancing structural and staining consistency between modalities. This framework outputs IHC-guided features from H&E images and employs a class-constrained contrastive loss for semantic consistency. Additionally, this paper integrates a bag feature augmentation strategy for WSI classifiers, enhancing feature diversity with the pre-trained model.

**Strengths:**

1.This paper proposes to use a diffusion model to generate IHC modal images and use the information in them to add additional semantics, which is somewhat innovative.
2.Extensive experiments were conducted on multiple datasets.

**Limitations:**

1.The information from IHC modality and HE modality is complementary, so it makes sense to combine both types of information for downstream task analysis. However, generating IHC modality from HE might be unreliable, because certain structures, such as the epithelia of lymphatic or blood vessels, are indeed indistinguishable in HE, and can only be accurately identified with CD-31 and D2-40 staining. I believe that using a small amount of proprietary data for pre-training, as mentioned in your experiment, may be unreliable.

2.To demonstrate the effectiveness of your approach, you can undertake an experiment similar to the example of lymphatic and blood vessels I described. Specifically, you can compare the identification and characterization of these structures using both HE and IHC modalities.

**Suitability:**

3

---

### Meta-Review · Area_Chair_KzK6 · 2024-06-30

**Recommendation:** Accept (Poster)
**Confidence:** 5

**Metareview:**

This paper proposed a multimodal denoising diffusion pre-training method for whole slide image classification. After rebuttal, the reviewers rated as 1 borderline accept, 1 borderline reject and 1 weak accept. The reviewers confirmed the merits of this paper and they also raised major concerns on the complexity of method and systematic experimental evaluation. After reconciling all the comments, I tend to suggest a decision of accept, but suggest the authors revise the manuscript by taking all the reviewers’ comments into consideration.